# The Chemokine Systems at the Crossroads of Inflammation and Energy Metabolism in the Development of Obesity

**DOI:** 10.3390/ijms222413528

**Published:** 2021-12-16

**Authors:** Pei-Chi Chan, Po-Shiuan Hsieh

**Affiliations:** 1National Defense Medical Center (NDMC), Department of Physiology & Biophysics, Taipei 114, Taiwan; peggychan7@gmail.com; 2Graduate Institute of Medical Science, NDMC, Taipei 114, Taiwan; 3Department of Medical Research, Tri-Service General Hospital, Taipei 114, Taiwan

**Keywords:** chemokines, inflammation, energy metabolism, obesity

## Abstract

Obesity is characterized as a complex and multifactorial excess accretion of adipose tissue accompanied with alterations in the immune and metabolic responses. Although the chemokine systems have been documented to be involved in the control of tissue inflammation and metabolism, the dual role of chemokines and chemokine receptors in the pathogenesis of the inflammatory milieu and dysregulated energy metabolism in obesity remains elusive. The objective of this review is to present an update on the link between chemokines and obesity-related inflammation and metabolism dysregulation under the light of recent knowledge, which may present important therapeutic targets that could control obesity-associated immune and metabolic disorders and chronic complications in the near future. In addition, the cellular and molecular mechanisms of chemokines and chemokine receptors including the potential effect of post-translational modification of chemokines in the regulation of inflammation and energy metabolism will be discussed in this review.

## 1. Introduction

The chemokines and seven-transmembrane G protein-coupled chemokine receptors have been documented to be the mediators of immune cell migration and response to the inflammatory reaction in injury and infection. Chemokines have also been reported to coordinate recruitment of various cell types in adipose tissues such as immune cells, adipocyte progenitors and endothelial cells in mice and humans [1]. The interaction between the chemokine system and these different cell types in brown and white adipose tissues (WAT) has been speculated to participate in the regulation of whole-body energy balance and glucose homeostasis [2,3,4,5,6]. However, the dynamic interaction between chemokines and different subsets of mesenchymal cells in the pathogenesis of the inflammatory milieu and dysregulated energy metabolism in obesity remains poorly understood.

Chemokines have been classified into CXC, CC, C and CX3C subfamilies in the base on the positioning of two amino-terminal cysteine residues. X in the chemokine could be any amino acid residue. Chemokines could bind with one or more chemokine receptors and also trigger different responses due to variant binding affinities to chemokine receptors for distinct chemokine ligands [7]. In addition, post-translational modification of chemokines by interaction with the extracellular matrix (ECM) and by binding to cell surfaces could significantly affect biological actions of chemokines and chemokine receptors. This review aims to present a significant updated study about the physiological and pathological roles of chemokines in energy metabolism and inflammation in the states of normal weight and obesity and the possible underlying mechanisms.

## 2. Chemokines in Obesity and Metabolic Syndrome

The increase in prevalence of obesity and associated cardiometabolic diseases has become a priority issue in health care systems worldwide and contributes to the global economic burden. Adipose tissue expansion in obesity consists of white and brown adipose tissues. WAT is the predominant site of energy storage and also serves as the endocrine organ. The adipocyte-derived hormones, cytokines and growth factors could regulate cell functions on local and/or systemic levels. Brown adipose tissue (BAT) is rich in mitochondria and capable of increasing energy expenditure and adaptive thermogenesis mainly via the activity of uncoupling protein 1 (UCP-1) [8]. There are some subcutaneous WATs possessing a subset of cells called beige adipocytes that can express high levels of UCP-1 upon chronic exposure to cold and β-adrenergic stimulation. Thereby, these cells are capable of elevated fuel oxidation and thermogenesis if needed. On the other hand, the chemokine levels in adipose tissue could be upregulated by the augmentation of inflammatory mediators in the obese adipose tissues [9,10,11]. However, it still remains elusive about the role of chemokines in deteriorating the development of adipose tissue dysfunction and insulin resistance in humans.

Previous reports have demonstrated that chemokines can facilitate the development of morbid obesity through its receptors to promote inflammatory leukocyte influx, especially pro-inflammatory monocytes, into hypertrophic adipose tissue. Circulating levels and tissue contents of the chemokines such as CXCL1, CXCL5, CXCL8, CXCL12, CCL2, CCL5, CCL7 and CCL19 are significantly increased in obesity. Abrogation of the signal pathways of CXCL14, CXCL5, CCL2, or their cognate chemokine receptors, alleviated obesity-associated metabolic abnormalities in high fat diet (HFD)-induced obese mice [1,12]. For instance, the study conducted with HFD-induced obese mice showed that deletion of *CCL2* or its receptor *CCR2* significantly attenuated inflammatory macrophage (ATM) infiltration and inflammatory reaction in adipose tissues [13,14,15] and consistent overexpression of CCL2 in adipose tissue increased inflammatory ATM content in WAT [16]. Nevertheless, deletion of *CCL2* gene did not affect obesity-associated monocyte influx into WAT [17]. On the other hand, previous reports have demonstrated that the increases of the expression CCL7 [18] in adipose tissues not only promotes migration of prostate cancer cells but also participates in a link between adipose tissue inflammation and insulin resistance in HFD-induced obese mice and the murine 3T3-F442A pre-adipocyte cell line. Increased adipose tissue expression of CCL19 in obese subjects has also been implicated to represent a pathological link between systemic low-grade inflammation and insulin resistance [19]. Taken together, these observations demonstrate that the chemokines/chemokine receptor-mediated signaling substantially contribute to the development of adipose tissue inflammation and subsequent metabolic disorders in obesity. However, the dualistic role of chemokines and chemokine receptors in the etiology of obesity-related inflammation and energy metabolism has been speculated on, but the data remain ambiguous.

## 3. The Involvement of Chemokines and Chemokine Receptors in the Development of Inflammation and Energy Metabolism

Metabolism homeostasis has been believed to be affected by cytokines secreted by immune cells as well as by adipocytes themselves in adipose tissue. Augmentation of tissue inflammation through adipocyte release of chemokines (e.g., CCL2 [13], CCL5 [20]) drives the alteration in leukocyte number and phenotype, thereby expanding the inflammatory environment within adipose tissue beds. The pro-inflammatory signals are also the important components of the thermogenic regulation in brown and beige adipocytes and contribute to their dysfunction in obesity by impairing energy expenditure and glucose uptake. In contrast, brown adipocytes release chemokines such as CXCL14- recruited, alternatively activated (M2) macrophages, and exerts its effects on promoting energy metabolism [21]. It is implicated that chemokines and chemokine receptors play a double-edged role in the progression of adipose tissue inflammation and energy metabolism, as shown in Figure 1.

### 3.1. CCL5

CCL5, also known as RANTES, has been documented to be involved in the pathogenesis of many inflammatory conditions in vivo. CCL5 could activate downstream signaling pathways such as nuclear factor NF-κB, and mitogen activated protein kinase (MAPK) pathways through its receptors, namely, CCR1, CCR3, and CCR5 [22,23,24]. In diabetes patients, CCL5 and CCR5 are upregulated in the peripheral blood [25,26] and they have also been well-documented as the upstream regulators of insulin resistance in obese individuals [27]. Besides, the increased serum levels of CCL5 in type 2 diabetic patients were closely related to postprandial hyperglycemia [28]. The onset of cardiovascular diseases in middle-aged females has been shown to be correlated to the CCL5 level in adipose tissue [29]. Augmented expression of CCL5 has been demonstrated to mediate the arrest and transmigration of monocytes/macrophages into the damaged site by binding with its receptor CCR5 [24,30,31]. CCL5 has also been reported to be involved into inflammatory reaction of WAT by recruiting blood monocytes and exerting antiapoptotic properties on WAT macrophages in obese subjects [20]. Moreover, Kim et al. reported that UV-induced CCL5 can suppress triglyceride synthesis in human adipocytes via downregulation of lipogenic enzymes. These observations suggest that CCL5 may participate in the pathogenesis of obesity-associated dysregulation of lipid metabolism and comorbidities [32].

On the other hand, CCL5 has also been reported to participate in the central control of energy metabolism in addition to their roles in mediating inflammation. Chou and colleagues showed that the hypothalamic CCL5/CCR5 signaling through the regulation of glucose uptake and AMPK activity might link to systemic insulin dependent glucose metabolism [33]. Furthermore, a clinical study evaluated the expression changes of CCL5 and CCR5 in obese humans under physical exercise. Their result showed that expression of CCL5 and CCR5 was higher in the subcutaneous adipose tissue of obese individuals compared with lean control and the elevated expression of CCL5 and CCR5 was significantly reversed through physical exercise [34]. Our recent study has demonstrated that obesity-induced augmentation of CCL5/CCR5 signaling suppressed adaptive thermogenesis by inhibiting AMPK-mediated lipolysis and oxidative metabolism in BAT to deteriorate the development of obesity [35]. However, the role of the CCL5-mediated signaling and whole-body energy metabolism still needs to be clarified further.

### 3.2. CX3CL1-CX3CR1 Signaling

CX3CL1, also known as fractalkine, is a chemokine with chemotactic activity for monocytes, T cells, and NK cells in the development of numerous inflammatory conditions in obesity-associated chronic complications such as atherosclerosis, insulin resistance and Type 2 diabetes [36,37,38]. For instance, the elevated CX3CL1 level has been reported in the blood of patients with type 2 diabetes and obesity [37]. CX3CL1 has been demonstrated to promote monocyte adhesion to human adipocytes as well as to evoke adipose tissue inflammation [37]. A study has shown that CX3CL1-mediating early recruitment of microglia induced by HFD might contribute to the induction of hypothalamic inflammatory response and subsequently the impairment of glucose tolerance and adiposity in experimental obesity [39]. Nevertheless, the role of the CX3CL1-CX3CR1 system in obesity-associated adipose tissue inflammation and insulin resistance remains controversial. For instance, *CX3CR1* deficient mice were protected against the development of HFD-induced obesity and WAT inflammation [40]. Deficiency of CX3CL1-CX3CR1 signaling resulted in the reduction of M2-polarized macrophage migration and an M1-dominant shift of macrophages within WAT. Moreover, mice lacking *CX3CR1* expression with HFD feeding displayed the reduced expression of proinflammatory cytokines and improved the profile of proteins involved in lipid metabolism and thermogenesis in BAT [41]. However, CX3CL1 administration in vivo has been reported to have a contradictory effect in diminishing glucose intolerance and insulin resistance [42]. Lee and colleagues also showed that *CX3CR1*^−/−^ mice developed glucose intolerance with diminished insulin secretion on both regular chow and HFD. *CX3CR1* deletion also promoted proinflammatory macrophage accumulation in adipose tissue and liver as well as insulin resistance [38]. Moreover, Nagashimada et al. reported that glucose intolerance, insulin resistance, and hepatic steatosis induced by HFD-induced obesity or leptin deficiency were exacerbated in *CX3CR1*^−/−^ mice. Thus, these contradictory observations implicate the dualistic role of the CX3CL1-CX3CR1 signaling in obesity-associated dysregulated glucose metabolism and adipose tissue inflammation. However, the underlying mechanism remains unclear.

### 3.3. CXCL12-CXCR4 Signaling

CXCL12, also known as stromal cell-derived factor 1 (SDF1), initially is a chemokine identified in bone marrow-derived stromal cells [43] and also an adipocyte-derived chemokine [44]. CXCL12 could regulate cell migration and survival through its receptor CXCR4 during cell development and tissue re-modeling [45]. Furthermore, blockage of CXCL12 action could reduce macrophage accumulation in adipose tissue via its receptor CXCR4 and eventually improved systemic insulin resistance in mice [44]. The CXCR4 in WAT has been reported to contribute to obesity-induced leukocyte recruitment, homeostasis, adipose tissue inflammation, and functional responses of adipocytes [46,47]. Moreover, the CXCL12-CXCR4 pathway has recently been speculated to be involved in the regulation of energy metabolism. For example, CXCL12-CXCR4 axis in adipose progenitors (APCs) has been demonstrated to contribute to ectopic fat deposition in high-fat fed mice through their migration to skeletal muscle to differentiate into adipocytes from subcutaneous fat [48]. Pharmacological antagonism of CXCR4 could prevent the effect of CXCL12 on APCs in subcutaneous fat and mimic the effects of overfeeding. A study conducted with *CXCR4* Loxp mouse and fatty acid-binding protein 4 -Cre mice has shown that *CXCR4* deficiency in either adipocyte or myeloid leukocyte could facilitate body weight gain and adiposity in HFD-induced obese mice through enhancing macrophage infiltration into the WAT and hypo-activity of brown adipocytes [46]. Accordingly, the UCP-1 mRNA and protein levels and oxygen consumption were significantly increased in the brown adipocytes treated with CXCL12 peptide [49]. In addition, the CXCL12-CXCR4 pathway has been further shown to participate in the control of adaptive thermogenesis and metabolic homeostasis through the activation of brown adipocytes in response to over-nutrition or intake of HFD [49]. In brief, the CXCL12-CXCR4 pathway could potentially affect several physiological processes related to the regulation of energy metabolism in adipose tissues.

### 3.4. CXCL14

The enhanced expression of CXCL14 was noted in WAT of HFD-fed obese mice. *CXCL14* deletion could reduce proinflammatory macrophage infiltration in WAT and improved insulin sensitivity in HFD-fed female mice [50]. CXCL14 deficiency could diminish HFD-induced hyperglycemia, hyperinsulinemia, hypo-adiponectinemia and hepatic steatosis in mice [50,51]. In addition, overexpression of CXCL14 in skeletal muscle restored obesity-induced insulin resistance in *CXCL14*^−/−^ mice [50,52]. It is implicated that CXCL14 is important not only in the regulation of insulin-dependent glucose uptake in skeletal muscles but also in control of whole-body glucose metabolism. Nevertheless, the in vitro study has also shown that CXCL14 has the pro-diabetogenic effects to inhibit glucose-stimulated insulin secretion in mouse islets [53].

Intriguingly, a research team found a causal link between CXCL14 protein and the regulation of energy metabolism. In their study, thermogenic stimuli led to the increase in CXCL14 levels and also the release of CXCL14 by BAT, which exhibits potential beneficial effects to obesity-associated metabolic diseases such as metabolic syndrome and type 2 diabetes. In addition, the authors have further demonstrated that CXCL14 might have a protective effect against insulin resistance by promoting the recruitment of alternatively activated M2 macrophages to adipose tissues and subsequently promote brown fat activity and white fat browning in subcutaneous adipose tissue [21]. However, there was a contradictory report that HFD-induced upregulation of CXCL14 could enhance insulin sensitivity by affecting adipocyte insulin signaling [54].

## 4. The Cellular and Molecular Mechanisms of Chemokines and Chemokine Receptors in the Regulation of Inflammation and Energy Metabolism

To elucidate the cellular and molecular mechanisms of chemokines and chemokine receptors in control of inflammation and energy metabolism under physiological and pathological conditions is crucial for developing therapeutic strategies for obesity-associated immune and metabolic disorders and chronic complications. However, the underlying mechanisms are not yet fully understood. In this session, we summarized the current understanding of the mechanism underlying the potential role of the chemokine systems on inflammation and energy metabolism. It might lead to future drug development for treatment of chemokine-related diseases, especially obesity-associated cardiometabolic disorders. On the other hand, there are two families of chemokine receptors in response to chemokine binding: conventional chemokine receptors (cCKRs) and atypical chemokine receptors (ACKRs). Chemokine-bound cCKRs typically transduce signals through pertussis toxin-sensitive GaiG-proteins and β-arrestins, ultimately leading to cell migration, adhesion and/or a variety of other biological responses. ACKRs are structurally related to cCKRs but do not couple to the G-protein-mediated signal transduction pathways activated by cCKRs [7,55]. They both could modulate biological effects of chemokines on tissue inflammation and energy metabolism.

### 4.1. The Cellular and Molecular Mechanisms of Chemokine-Mediated Tissue Inflammation

Tissue inflammation is attributed to local immune, vascular and inflammatory cell responses to acute infection or injury and also exhibits in chronic inflammatory diseases such as autoimmune diseases [56,57,58], allergy [59], neuron degeneration disease [60], chronic inflammatory disease [61], cardiovascular disease [62], cancer [63] and morbid obesity [64]. Chemokines are multifunctional mediators mainly responsible for leukocyte recruitment to inflamed organ systems and control the innate immune cell trafficking between the bone marrow, blood, and peripheral tissues during inflammation through chemokine receptors. These inflammatory chemokine profiles typically include cCKRs such as CCR1, CCR2, CCR3, CCR5, CXCR1, CXCR2, and CXCR3 and ACKRs such as ACKR1, and ACKR2. However, the precise chemokine profile in a given tissue is depend on the exact nature of the chemokine network in the affected individual and also the received stimuli.

For instance, glucotoxicity that occurs in both type 1 and type 2 diabetes could facilitate the synthesis and release of chemokines by activating pancreatic β-cell NLRP3 inflammasomes [65] and the downstream inflammatory signal pathways such as IL1R and NF-κB [66]. During hepatic inflammation, both hepatocytes, stellate cells, sinusoidal endothelial cells and infiltrating leukocytes are the source of CXCR3 ligands. The expression of CXCR3 ligands requires stimulation with IFN-γ and TNF-α, which are released by activated hepatic macrophages, Kupffer cells and the initial wave of infiltrating innate immune cells [67,68]. Growing evidence has revealed that obesity is associated with immune response involving chemokines secreted by immune cells, such as CCL2, and CCL5. These proinflammatory chemokines could aggravate systemic inflammation by recruiting pro-inflammatory M1 macrophages, and contributes to the onset of obesity-associated comorbidities [12,13,20]. Moreover, Th1 and CD8 T cells are elevated in adipose tissues of obese subjects, which are major IFNγ-expressing cells that accumulates in obesity. IFN-γ stimulates the expression of pro-inflammatory chemokines in adipocytes as well as the M1 polarization of macrophages [69,70]. CC chemokines, CCL2, CCL3, and CCL5, elevated in the joints of patients with rheumatoid arthritis, coincide with the recruitment of monocytes and T cells into synovial tissues. CXCL10 and CXCL8 chemokines were also elevated in the plasma of patients with active rheumatoid arthritis similar to the Th1 associated proinflammatory cytokines TNFα, and IL-6 [71,72].

### 4.2. The Cellular and Molecular Mechanisms of Chemokine and Chemokine Effects on Energy Metabolism

The role of CCL2 in regulating AMPK activity has been demonstrated. Inflammatory stimuli increase the production of CCL2 and, subsequently, decrease the activation of AMPK, resulting in increased mTOR activity and, ultimately, altered energy metabolism [73]. Adipocyte CCL19 inhibited AMPKα through activating extracellular signal-regulated kinase (ERK)1/2, resulting in impaired lipid metabolism and energy regulation [74].

Chemokine-mediated signaling such as CCL2 can activate the phosphoinositide 3-kinases (PI3Ks) pathway [73,75]. The PI3Ks family of proteins such as lipid kinases and its lipid product, phosphatidylinositol triphosphate, have been shown to be involved in different cell functions, including regulating proliferation, cell polarity, cell survival and chemotaxis [76,77,78]. In addition, PI3K/AKT signaling regulates glucose metabolism. Notably, chemokines and chemokine receptors such as CXCL8 and CXCR4 have been reported to activate the PI3K/AKT signaling through chemokine receptors in osteosarcoma and pancreatic cancer cells [79,80]. AKT can directly stimulate glucose transporter (GLUT) 1 and induces translocation of GLUT 4 to cell membrane to facilitate glucose uptake into mammalian cells. Phosphofructokinase 2 is a rate-determining enzyme of glycolysis can also be activated by AKT [81].

In the tumor microenvironment, CCL5 directly interacts with CCR5-expressing breast cancer cells to promote anabolic metabolic events leading to enhanced cell proliferation and tumorigenesis. On the other hand, CCL5 has been reported to regulate T cell rolling, adhesion, and transmigration efficiency to promote an effective immune response [82]. CCL5-mediated glucose uptake and ATP accumulation to meet the energy demands of chemotaxis in activated T cells [83]. In addition, it has also been demonstrated that CCL5 participates in macrophage chemotaxis. CCL5 enhanced RAW264.7 macrophage migration and glucose uptake through up-regulated membrane GLUT1 expression as well as phosphorylation of AKT which has prolonged effect on phosphorylation of AMPK [84]. However, the pathological and physiological roles of CCL5-mediated signaling in obesity-related dysregulation of energy metabolism remain mostly unclear.

The CXCL12-CXCR4 pathway elicits brown adipocyte activity and affects nutrient metabolism through upregulation in p38 and ERK levels under energy overnutrition [49]. A novel regulatory factor, CXCL14, secreted by BAT in response to cold exposure, has been shown to increase energy expenditure through promoting M2 macrophages recruitment and browning of WAT and, eventually, could counterbalance the detrimental effect of energy excess under the state of energy overnutrition [21].

### 4.3. The Potential Effects of Post-Translational Modification of Chemokines on Inflammation and Energy Dysregulation in the Development of Obesity

Chemokines are profoundly affected by post-translational modification, which regulates chemokine localization and abundance by interaction with the ECM and by binding to cell surfaces. The complex physical interactions existing in the chemokine network could significantly affect biological actions of chemokines and chemokine receptors. The post-translational modification of chemokines such as citrullination [85,86,87], nitration/nitrosylation [88,89,90], and cleavage by matrix metalloproteinases (MMPs), cathepsins, thrombin, plasmin, elastase, the dipeptidyl peptidase CD26, and other proteases [91,92,93] can substantially modify chemokine activity. For instance, nitration of tyrosine residues in CCL2 by reactive nitrogen species reduces the ability of this chemokine to attract monocytes through its receptor CCR2 [88], which might be involved in obesity-induced adipose tissue inflammation [15]. Proteases such as dipeptidyl peptidase-4 (DPP4, also called CD26, cluster of differentiation 26) and MMPs are key chemokine regulators. DPP4-mediated trimming of two amino acids off a chemokine’s N terminus can change receptor specificity, substantially alter chemokine receptor affinity or convert agonists into antagonists [91]. Many MMPs have similar effects as DPP4 on selected chemokines. MMP-mediated N-terminal trimming could enhance the activity of CXCR2 ligands, and of CC chemokines with extended N-termini (i.e., CCL6, CCL9, CCL23), while CXCL12 is inactivated by MMPs and CCR2 ligands are converted into receptor antagonists [94].

Glycosaminoglycans (GAGs) are long unbranched polysaccharides which are composed of repeating disaccharide units and also referred to as mucopolysaccharides due to their viscous and lubricating properties. Their ability to bind chemokines influences chemokine/receptor interactions and also the half-life of chemokines in vivo [94,95]. In addition, TNF-stimulated gene-6, a multifunctional protein secreted in response to pro-inflammatory stimuli by monocytes and endothelial cells, has been shown to mediate anti-inflammatory effects by inhibiting chemokine/GAG interactions to alter chemokine distribution and function [96,97]. On the other hand, chemokines can act as monomers and also form homodimers and heterodimers. They can contain one or more chemokine species via interactions with GAGs [98,99,100,101,102]. For example, human CXCL4 and CCL5 can heterodimerize or oligomerize with over 20 other chemokines from CC, CXC, and XC subfamilies, respectively [102]. Oligomerization has been demonstrated to involve into the mechanism underlying individual chemokines or mixtures of them in control leukocyte responses [78,97,101,102,103]. Thereby, blockage of specific interactions as mentioned above may be the therapeutic targets to treat chemokine-induced immune and metabolic disorders in obesity.

## 5. Modulation of the Chemokine/Chemokine Receptor Axis as a Novel Approach for Treatment of Morbid Obesity

The complex chemokine network composed of a large number of interacting ligands, receptors and regulatory proteins engaged in diverse cellular processes in different given organs may significantly contribute to obesity-associated adipose tissue inflammation and whole-body energy dysregulation. Accordingly, the importance of chemokines and chemokine receptors in obesity-related adipose tissue inflammation and subsequent development of insulin resistance has been well-documented. Recently, some chemokines and their receptors, as discussed above, have been speculated to have diverse effects on regulation of immune reaction and energy metabolism as shown in Table 1. For instance, C3XCL1 could promote inflammatory reactions but suppress energy expenditure in BAT and WAT. On the other hand, CXCL12 and CXCL14 could facilitate energy expenditure in BAT but suppress energy metabolism and also provoke inflammatory reaction in WAT. Thereby, the chemokine system could become the potential therapeutic target of obesity-associated immune-metabolic abnormalities to prevent the development of obesity-related chronic complications such as metabolic syndrome, type 2 diabetes and cardiovascular diseases [104].

For example, administration of TAK-779, a dual inhibitor of chemokine receptors CCR2 and CCR5, has been demonstrated to suppress obesity-related accumulation of leukocytes in the adipose tissue and be an efficient way to prevent insulin resistance and beta-cell dysfunction [105]. Compounds such as INCB8761/PF-413630 targeting adipose tissue in the treatment of obesity-associated diabetes, are currently pursued in human clinical trials and appear to be a new series of CCR2 antagonists that are orally bioavailable [106]. In brief, although clinical translation is still under slow progression, drugs targeting cCKRs in the development of obesity could be the potential therapeutic drug targets to made it to the clinic in the near future.

## Figures and Tables

**Figure 1 ijms-22-13528-f001:**
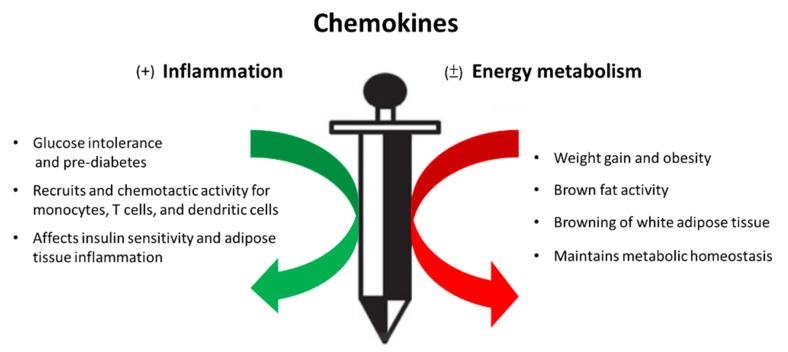
Chemokines, a double-edged sword in the inflammatory response and energy metabolism during the process of metabolic syndrome.

**Table 1 ijms-22-13528-t001:** The summary of the functional roles of adipocyte-derived chemokines.

Chemokines	Chemokine Receptors	Brown/Beige Adipocytes			White Adipocytes			References
		Inflammation		Energy Expenditure	Inflammation		Energy Expenditure	
		Pro	Anti		Pro	Anti		
CCL5	CCR5	↑	↓	↓	↑	↓	↓	[30,31,32,33,34,35]
C3XCL1	CX3CR1	↑	↓	↓	↑	↓	↓	[37,38,39,40,41,42]
CXCL12	CXCR4	NA	NA	↑	↑	↓	↓	[44,46,47,48,49]
CXCL14	NA	NA	NA	↑	↑	↓	↓	[50,51,52,53,54]

## Data Availability

All the data are presented in the manuscript.

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
