# Peer review of "The Chemokine Systems at the Crossroads of Inflammation and Energy Metabolism in the Development of Obesity"

_ijms, 2021, doi:10.3390/ijms222413528_

Round 1
Reviewer 1 Report
This review aims to present the link between chemokines and obesity-related inflammation and metabolic dysregulation, which could present important therapeutic targets that could control obesity-associated immune and metabolic disorders and chronic complications in the near future.
The review is well structured, nevertheless the authors should refer to more recent publications on the subject (only a few references are from 2020-2021).
minor points:
- line 139, ... replace "in patients of type 2 diabetes .." with "in patients with type 2 diabetes".
- line 143, use the abbreviation HFD after having entered the definition in full. Check all abbreviations.
- line 252, why do the authors list MCP-1 and CCL2 as if they were two different markers?
Author Response
Responses to reviewer #1
- The review is well structured, nevertheless the authors should refer to more recent publications on the subject (only a few references are from 2020-2021). Answer: We has updated the references as suggested.
minor points:
(1) - line 139, ... replace "in patients of type 2 diabetes .." with "in patients with type 2 diabetes". Answer: Revised as suggested.
(2) - line 143, use the abbreviation HFD after having entered the definition in full. Check all abbreviations. Answer: Revised as suggested.
(3) - line 252, why do the authors list MCP-1 and CCL2 as if they were two different markers? Answer: Thanks for the comment. MCP-1 in line 287 has been deleted.

Reviewer 2 Report
Dear Authors,
The manuscript presents an interesting topic concerning the chemokine systems at the crossroads of inflammation and energy metabolism in the development of obesity. The authors aimed to review the physiological and pathological roles of chemokines in energy metabolism and inflammation in the states of normal weight and obesity and the possible underlying mechanisms.
I have some considerations regarding missing elements in the manuscript.
- There are some shortcuts without explanation in the manuscript. Please, explain the shortcuts in the proper order in the text.
- Authors should not explain shortcuts again in the manuscript's text. Please, check the text carefully.
- Please describe the "genetic inactivation" method while citing a study.
- Lines 38-40, the word “different” is used 3 times
- The source of chemokines expression should be localized while described and pointed if it concerns transcript or protein level.
- Please use the proper citations style in the References section. For example, some references contain only authors' first names without surnames.
- Figure 1 and Table 1 should not situate at the manuscript's end. Please, move them to a more appropriate position.
- Table 1. has typo mistakes, and authors should carefully rearrange the data. Authors should consider showing data in all presented columns, not leave columns empty.
- The manuscript may benefit from an additional schema, figures regarding the presented topic. Color figures are more attractive for online readers, especially in literature reviews.
- Metabolic syndrome is defined as some components (at least three) existing together and may not involve obesity. Thus, please, precise the target clinical condition of interest while describing actions of chemokines.
- Please provide and explain the exact names of animal models mentioned in the manuscript, the type of diet used in the induction of models, exact names of cell lines used in the described experiments. Also, the authors should explain the shortcuts used.
- The reference "Keophiphath et al." is difficult to find in the References section.
- The potential therapeutic target of the chemokine system in obesity (and metabolic syndrome) should be extended in the last section of the manuscript to examples from published studies.
14. Authors should improve the English language.
Author Response
Responses to reviwer #2
- There are some shortcuts without explanation in the manuscript. Please, explain the shortcuts in the proper order in the text.
Answer: Revised as suggested.
- Authors should not explain shortcuts again in the manuscript's text. Please, check the text carefully.
Answer: Revised as suggested.
- Please describe the "genetic inactivation" method while citing a study.
Answer: Thanks for the comment. The related sentence has been reworded by
using deletion of CCL2 gene instead of genetic inactivation of CCL2 in
line 104, page 3.
- Lines 38-40, the word “different” is used 3 times
Answer: The related sentence has been reworded as “Chemokines could
bind with one or more chemokine receptors and also trigger different
responses due to variant binding affinities to chemokine receptors for distinct
chemokine ligands [7].”. in line 68-70, page 2.
- The source of chemokines expression should be localized while described and pointed if it concerns transcript or protein level.
Answer: Revised as suggested.
- Please use the proper citations style in the References section. For example, some references contain only authors' first names without surnames.
Answer: We have re-checked all the references and revised as suggested.
- Figure 1 and Table 1 should not situate at the manuscript's end. Please, move them to a more appropriate position.
Answer: Revised as suggested.
- Table 1. has typo mistakes, and authors should carefully rearrange the data. Authors should consider showing data in all presented columns, not leave columns empty.
Answer: Revised as suggested.
- The manuscript may benefit from an additional schema, figures regarding the presented topic. Color figures are more attractive for online readers, especially in literature reviews.
Answer: Thanks for the comments from reviewer. Accordingly, the figure 1 has
been revised as suggested. Nevertheless, it would be difficult to draw an
additional meaningful schema or figures to date since only a few chemokines and chemokine receptors have been studied and most of the role and mechanism of chemokines remain unclear.
- Metabolic syndrome is defined as some components (at least three) existing together and may not involve obesity. Thus, please, precise the target clinical condition of interest while describing actions of chemokines.
Answer: Thanks for reviewer’s comment. The metabolic syndrome consists of clustering of abdominal obesity, dyslipidemia, hyperglycemia and hypertension, is a major public health challenge. Abdominal obesity is the most frequently observed component of metabolic syndrome. However, as pointed by reviewer, it should be more cautiously to describe the actions of chemokines on obesity and metabolic syndrome in the text. Accordingly, we have scrutinized through the text and revised the related statement as suggested.
- Please provide and explain the exact names of animal models mentioned in the manuscript, the type of diet used in the induction of models, exact names of cell lines used in the described experiments. Also, the authors should explain the shortcuts used.
Answer: Revised as suggested. Thanks for the comments.
- The reference "Keophiphath et al." is difficult to find in the References section.
Answer: The related sentence has been reworded as suggested in line 145-148, page 4.
- The potential therapeutic target of the chemokine system in obesity (and metabolic syndrome) should be extended in the last section of the manuscript to examples from published studies.
Answer: Thanks for the helpful advise. The related sentence has been revised as
suggested in the last section.
- Authors should improve the English language. Answer: The manuscript has been scrutinized and improved the English writing as possible.
